# Treating Parkinson’s Disease with Human Bone Marrow Mesenchymal Stem Cell Secretome: A Translational Investigation Using Human Brain Organoids and Different Routes of In Vivo Administration

**DOI:** 10.3390/cells12212565

**Published:** 2023-11-02

**Authors:** Bárbara Mendes-Pinheiro, Jonas Campos, Ana Marote, Carina Soares-Cunha, Sarah L. Nickels, Anna S. Monzel, Jorge R. Cibrão, Eduardo Loureiro-Campos, Sofia C. Serra, Sandra Barata-Antunes, Sara Duarte-Silva, Luísa Pinto, Jens C. Schwamborn, António J. Salgado

**Affiliations:** 1Life and Health Sciences Research Institute (ICVS), School of Medicine, University of Minho, Campus de Gualtar, 4710-057 Braga, Portugal; 2ICVS/3B’s—PT Government Associate Laboratory, 4805-017 Guimarães, Portugal; 3Luxembourg Centre for Systems and Biomedicine (LCSB), University of Luxembourg, L-4367 Belvaux, Luxembourg

**Keywords:** mesenchymal stem cells, brain organoids, stem cell secretome, Parkinson’s disease, pre-clinical study, neurodegeneration

## Abstract

Parkinson’s disease (PD) is the most common movement disorder, characterized by the progressive loss of dopaminergic neurons from the nigrostriatal system. Currently, there is no treatment that retards disease progression or reverses damage prior to the time of clinical diagnosis. Mesenchymal stem cells (MSCs) are one of the most extensively studied cell sources for regenerative medicine applications, particularly due to the release of soluble factors and vesicles, known as secretome. The main goal of this work was to address the therapeutic potential of the secretome collected from bone-marrow-derived MSCs (BM-MSCs) using different models of the disease. Firstly, we took advantage of an optimized human midbrain-specific organoid system to model PD in vitro using a neurotoxin-induced model through 6-hydroxydopamine (6-OHDA) exposure. In vivo, we evaluated the effects of BM-MSC secretome comparing two different routes of secretome administration: intracerebral injections (a two-site single administration) against multiple systemic administration. The secretome of BM-MSCs was able to protect from dopaminergic neuronal loss, these effects being more evident in vivo. The BM-MSC secretome led to motor function recovery and dopaminergic loss protection; however, multiple systemic administrations resulted in larger therapeutic effects, making this result extremely relevant for potential future clinical applications.

## 1. Introduction

Parkinson’s disease (PD) is recognized as a multi-systemic neurodegenerative disorder with motor and non-motor features resulting from a severely perturbed neurotransmitter system. Neuronal loss in the substantia nigra pars compacta (SNpc), which causes striatal dopamine (DA) deficiency, and intracellular inclusions containing aggregates of α-synuclein are the main neuropathological hallmarks of PD [1,2]. Multiple other cell types of the central and peripheral autonomic nervous system are also affected, most likely from the beginning of the disease [1]. Motor dysfunctions largely depend on the denervation of the nigrostriatal pathway. Therefore, the treatment of PD is anchored on DA pharmacological substitution in addition to non-dopaminergic approaches to address both motor and non-motor symptoms, as well as deep brain stimulation for those developing levodopa-related motor complications [3]. Although clinical improvements can be achieved, most of the current approaches lead to the development of serious side effects and are only effective in a subset of affected individuals for a restricted time window [4]. Moreover, none of them can ameliorate the neuropathological processes leading to progressive cell death. Mesenchymal stem cell (MSC)-based approaches have been suggested as promising therapeutic alternatives for neurodegenerative disorders due to their inherent characteristics [5]. MSCs are a heterogeneous group of multipotent cells, known for their non-tumorigenicity and immunomodulatory properties, that can be isolated from numerous adult tissues being able to grow at numbers required for clinical usage under good-manufacturing practices (GMP) compliance without major ethical issues [6]. The beneficial effects of MSCs were primarily attributed to their ability to engraft and differentiate into multiple tissue types, including dopaminergic neurons. However, recent evidence has revealed that implanted cells do not survive for long periods and that the benefits of MSC therapy are associated with their paracrine activity [6,7]. In recent years, compelling evidence has brought attention to the array of bioactive molecules produced by MSCs, collectively known as secretome [6,8,9]. Indeed, different studies have demonstrated that secretome-derived products from MSCs have neuroregulatory actions on key pathological processes that are associated with basic homeostatic functions, such as cell differentiation and proliferation, angiogenesis and vasculogenesis, inflammation, and oxidative stress [8,10]. Nevertheless, the design of innovative disease-modifying therapies for PD requires the use of experimental models that better recapitulate the pathophysiological mechanisms underlying dopaminergic degeneration. Therefore, the use of in vitro models provides unique opportunities for assessing molecular aspects of neurodegeneration and for the identification of novel therapeutic strategies [11]. The discovery of induced pluripotent stem cells (iPSCs) revolutionized the stem cell field, allowing the use of human models to study a variety of diseases, including PD [12]. Traditionally, iPSCs have been cultured using 2D techniques that do not fully represent complex tissues like the human brain. In recent years, the generation of 3D in vitro models of complete organs—the so-called organoids—emerged as a promising disease model [13]. These complex in vitro systems mimic organ architecture and function and have been shown to model different neurological disorders [14]. Concerning PD, only a few studies used human midbrain-specific organoids (hMOS); however, they proved that organoids can be used to model aspects of PD-related neuronal pathology [15,16,17]. Furthermore, the evaluation of potential new treatments for PD employing brain organoids is still in its infancy, with only a few published studies demonstrating the need for further investigations [18,19,20,21].

For the past decade, our lab has shown that intracerebral injections of bone-marrow-derived MSCs (BM-MSCs) secretome were able to protect dopaminergic neurons in injured areas and induce partial motor function recovery in a complete lesion of a 6-hyrdoxydopamine (6-OHDA) rat model of PD [22,23]. Despite these promising outcomes, brain local infusion approaches are invasive and have disadvantages such as high cost, risk of infection, and a limited delivery brain area [24]. On the other hand, the use of noninvasive systems for MSC secretome delivery, such as systemic administration through intravenous injections, could avoid these problems and may be more easily implemented in clinical settings. Nevertheless, a straight comparison between these two modes of administration in PD-injured animals has never been conducted. Therefore, in the present study, we used an optimized midbrain organoid system as an in vitro paradigm to model dopaminergic degeneration and studied the impacts of BM-MSCs secretome. Furthermore, the efficacy of intracerebral injections (single administration in two sites) with multiple systemic administrations of BM-MSC secretome was compared using a unilateral intrastriatal 6-OHDA PD mouse model.

## 2. Materials and Methods

### 2.1. BM-MSC Culture, Secretome Collection and Concentration

Human bone marrow-derived MSCs (BM-MSCs; Lonza, Basel, Switzerland) were thawed and seeded into T75 cm^2^ tissue culture flasks and expanded in MesenCult™-ACF Plus Medium (STEMCELL Technologies, Vancouver, BC, Canada) for the in vitro study and expanded in Minimum Essential Medium α (α-MEM; ThermoFisher Scientifc, Waltham, MA, USA) supplemented with 5% human platelet lysate (HPL) (PL Biosciences, Aachen, Germany), 1% Pen-Strep (Gibco, Grand Island, NY, USA), and 1.8 mM heparin (B.Braun, Melsungen, Germany) for the in vivo experiments. The medium was renewed every 3 days and upon confluence, cells were dissociated using Animal Component-Free Cell Dissociation Kit (STEMCELL Technologies) for BM-MSCs growing in MesenCult medium according to the manufacturer’s instructions or dissociated using 0.05% trypsin-EDTA solution (ThermoFisher Scientific, Waltham, MA, USA) for BM-MSCs growing in α-MEM 5% hPL. The BM-MSCs used in this work were phenotypically characterized by our laboratory, being positive for the standard MSC markers CD73, CD90, and CD105, and negative for CD45 and HLA-DR. The functional characterization was also evaluated by chondrogenic, osteogenic, and adipogenic differentiation [25].

For all experiments, the secretome of BM-MSCs was collected under the form of conditioned medium (CM) in passage 8 (P8) according to protocols already established in our laboratory [21,22]. Briefly, 5000 cells/cm^2^ were seeded and expanded for 72 h in the respective growth media. On the conditioning day, cells were washed 5 times with PBS without Ca^2+^/Mg^2+^ (Invitrogen, Waltham, MA, USA) and incubated with Neurobasal A medium (ThermoFisher Scientific) supplemented with 1% kanamycin (Life Technologies, Carlsbad, CA, USA). After 24 h, the media containing the factors secreted by BM-MSCs were collected and centrifuged at 1200 rpm for 10 min at 4 °C to remove any cell debris. For both studies (in vitro and in vivo), the secretome was 100× concentrated by centrifugation using a 5 kDa cut-off concentrator (Vivaspin, GE Healthcare, Hatfield, UK), and frozen at −80 °C until used.

### 2.2. Neuroepithelial Stem Cell Derivation and Generation of Midbrain Organoids

All work with human stem cells was carried out after approval of the national ethics board, Comité National d’Ethique de Recherche (CNER), under the approval numbers 201305/04 and 201901/01 [17].

iPSCs were generated as described in the study of Reinhardt et al., [26] or bought from Coriel, and derivation of human ventralized NESCs (hvNESCs) from iPSCs via embryonic body formation is described in detail in the study of Smits et al. [16]. Three different WT lines were used, namely K7 (derived from fibroblasts from an 81-year-old female), T12 (derived from a 53-year-old female), and COR (derived from a 55-year-old male) (Appendix A). Briefly, hvNESCs were maintained for up to 15 passages (after their generation from iPSCs) under maintenance medium, split at a confluency of 80% using Accutase (ThermoFisher Scientific, USA), and seeded 600,00 cells per well on Geltrex (ThermoFisher Scientific) coated 6-well plates. N2B27 maintenance media consisted of equal parts of DMEM/F12 (ThermoFisher Scientific) and Neurobasal (ThermoFisher Scientific) supplemented with L-glutamine (1:100; ThermoFisher Scientific), penicillin-streptomycin (Pen-Strep, 1:100; ThermoFisher Scientific), N2 (1:200; ThermoFisher Scientific), and B27–Vitamin A (1:100; ThermoFisher Scientific). N2B27 was freshly supplemented with 3 μM CHIR 99021 (CHIR; Axon Medchem, Groningen, Netherlands), 2.5 μM SB-431542 (SB; Ascent Scientific, London, UK), 100 nM LDN-193189 (LDN; Sigma, St. Louis, MO, USA), 0.5 μM SAG (Merck, Rahway, NJ, USA), and 200 μM ascorbic acid (AA; Sigma). All the antibodies used to characterize hvNESCs can be found in the Appendix A.

To generate the hMOS from hvNESCs, 9000 cells were seeded per well into ultra-low attachment 96-well round-bottom plates (Corning, Corning, NY, USA) and kept under maintenance conditions for colony formation for 8 days. Patterning I medium consists of maintenance medium without SB and LDN. For patterning II medium, the concentration of CHIR was reduced to 0.7 μM. On day 8, the 3D colonies were embedded into geltrex droplets [27] and cultivated into non-treated 24-well tissue culture plates (CELLTREAT, Pepperell, MA, USA) in N2B27 differentiation media (one organoid was kept per well). Differentiation media consist in N2B27 media supplemented with 200 μM AA, 500 μM dibutyryl cAMP (db-cAMP, Sigma), 10 ng/mL brain-derived neurotrophic factor (BDNF; Peprotech, London, UK), 10 ng/mL glial cell-line-derived neurotrophic factor (GDNF; Peprotech), 1 ng/mL transforming growth factor β3 (TGF-β3; Peprotech), 10 μM DAPT (R&D Systems, Minneapolis, MN, USA), and 2.5 ng/mL Activin A (Peprotech). The organoids were placed on an orbital shaker rotating at 80 rpm, and media was changed every third day for 22 days as depicted in Figure 1.

### 2.3. 6-OHDA Injury and Secretome Treatments In Vitro

6-OHDA was used to induce dopaminergic cell death. 6-OHDA treatment was performed on day 30 of culture. At this time-point, robust neuronal differentiation has been achieved in the hMO cultures. 6-OHDA hydrobromide (100 μM, calculated from freebase weight; 162957, Sigma) was prepared on ice in the dark and dissolved in N2 media without growth factors (DMEM/F12 with 1% N2, 1% PenStrep, and 1% Glutamax (Gibco, USA). B27 was not added to the media, to avoid reducing the effect of 6-OHDA, due to its antioxidant capacities. Because of the instability and rapid auto-oxidation of 6-OHDA, a double-treatment was performed (two consecutive days for 24 h). After neurotoxin exposure, 6-OHDA was removed, and the wells were incubated with the following conditions for 6 days (with one media change, Figure 1A): differentiation medium N2B27 (Control+), Neurobasal A medium (basal medium without supplementation; Control−), and BM-MSCs conditioned medium (Secretome).

### 2.4. hMO Immunofluorescence Staining

At the end of experiment, hMOs were fixed with 4% paraformaldehyde (PFA) overnight at RT and washed 3× with phosphate-buffered saline (PBS) for 15 min. Then, hMOs were embedded in 3% low-melting point agarose in PBS and incubated for 20 min at 40 °C, followed by 30 min incubation at RT. Next, 80 µm sections were cut using a vibratome (Leica VT1000s, Wetzlar, Germany), and sections were separated into “border” and “center” to provide an adequate spatial representation of the hMOs. The sections were permeabilized and blocked with 0.5% Triton X-100, 2.5% normal goat serum (Invitrogen), 2.5% BSA, and 0.1% sodium azide. Sections were incubated on a shaker for 48 h at 4 °C with primary antibodies in the blocking buffer containing 0.1% Triton X-100 at the following dilutions: anti-TH from rabbit (1:1000; ab112, Abcam, Cambridge, UK), anti-TUJ1 (hybrodoma clone to produce antibodies against beta-III tubulin) from mouse (1:600; 801201, BioLegend, San Diego, CA, USA), anti-MAP2 from chicken (1:1000; ab5392, Abcam) and anti-Dopamine from rabbit (1:500; IS1005, ImmuSmol, Bordeaux, France). After incubation with the primary antibodies, sections were washed three times in 0.01% Triton X-100 and incubated with the secondary antibodies (1:1000) including a Hoechst 33342 counterstaining for nuclei in blocking buffer with 0.01% Triton X-100. All secondary antibodies (Invitrogen) were conjugated to Alexa Fluor fluorochromes with appropriate excitation/emission spectra combinations. Fluorescence images were acquired on an Operetta confocal microscope (Perkin Elmer, Waltham, MA, USA) with a 20× objective (16–20 area scans, 25 z-planes).

### 2.5. Image Analysis

Immunofluorescence 3D images of hMOs were analyzed in Matlab (Version 2017b, Mathworks, Natick, MA, USA) as previously described in detail [15]. The in-house developed image analysis algorithms automate the segmentation of nuclei and neurons, with structure-specific feature extraction. High-content image analysis workflow and the features that are possible to extract are represented in the Appendix A.

### 2.6. Animals

Ten-weeks old male and female C57/BL6J, weighing 18–28 g at the beginning of the experiment were used according to local regulations on animal care and experimentation (European Union Directive 2010/63/EU), and with consent from the Portuguese national authority for animal research, Direção Geral de Alimentação e Veterinária (ID: DGAV 005,454 23/04/2025, Lisbon, Portugal) and Ethical Subcommittee in Life and Health Sciences (SECVS; ID: SECVS-142/2016, University of Minho, Braga, Portugal). Animals were housed in groups of 5–6 animals, in a temperature- and humidity-controlled room, maintained on 12:12 h dark/light cycles, with access to food and water ad libitum.

### 2.7. Lesion Surgery and Post-Operative Care

Mice were anesthetized by intraperitoneal injection of ketamine (75 mg/kg) plus medetomidine (1 mg/kg) diluted in 0.9% NaCl, and placed on a stereotaxic apparatus (Stoelting, Kiel, WI, USA). The lesioned group (*n* = 75) was injected unilaterally with 6 μg/μL of 6-OHDA hydrochloride (H4381, calculated from freebase weight; Sigma, USA; dissolved in of 0.9% NaCl in 0.2 mg/mL ascorbic acid) into the dorsolateral striatum (2 × 1.5 μL) at a rate of 300 nl/min, at the following coordinates (relative to bregma, in mm): (i) AP = +1.0, ML = 2.1, DV = −2.9; (ii) AP = +0.3, ML = −2.3, DV = −2.9. After each injection, the needle was left in place for 5 min for diffusion and to avoid any backflow. The sham group (*n* = 15) was injected in the same conditions with the neurotoxin vehicle. Anesthesia was reversed using atipamezole (1 mg/mL).

The animals were monitored twice a day for 14 days after surgery, and mice that reached the established humane endpoints [28] were euthanized. During the recovery, animals were housed in a warmed room and post-operative care included the administration of nutrients and fluids in addition to post-operative analgesia, as previously described in detail [29].

### 2.8. Behavioral Assessment, Secretome Treatments and Experimental Design

Three weeks after 6-OHDA injections, animals were submitted to an initial behavioral analysis to assess the extent of motor deficits using the beam balance walk test (round beam 11 mm), the motor swimming test, the pole test, and the rotameter test using apomorphine. All behavioral tests were performed as previously described in detail [29]. Body weight was also registered every week until the end of the experiments (Appendix A). After 2 weeks, 6-OHDA-lesioned animals were then divided with a block randomization strategy as follows: 6-OHDA (IC Vehicle) intracerebral (IC) injections of Neurobasal A medium; 6-OHDA (IC Secretome)—IC injections of BM-MSC CM; 6-OHDA (IV Vehicle)—intravenous (IV) injections of Neurobasal A medium; 6-OHDA (IV Secretome)—co IV injections of BM-MSC CM. Regarding IC injections, animals received 2 × 1.5 μL of BM-MSC secretome and the respective vehicle in the dorsal striatum (same coordinates used in lesion surgeries) and 1 μL in SNpc (coordinates related to bregma, in mm: AP = −3.0, ML = 1.2, DV = −4.5). The protein concentration of the injected 100× concentrated secretomes was 318 µg/mL. The total amount of protein injected per animal from the IC Secretome group was 1.27 µg. After surgery, all animals were treated with the analgesic buprenorphine at 0.05 mg/kg (Bupaq; Richter Pharma, Oberösterreich, Austria). The other two groups received IV injections (3 injections in the same week of IC surgeries, and then once a week for 7 weeks) of 100 μL (total amount of protein/IV injection: 31.8 µg) in the tail vein. Motor behavioral analysis was performed 1, 4, and 7 weeks after treatments, and mice were posteriorly sacrificed to proceed with brain tissue analysis.

### 2.9. Histological Assessment

To further evaluate the degree of dopaminergic preservation, immunohistochemical staining for tyrosine hydroxylase (TH) was performed. For that, at the end of behavioral analysis, animals were deeply anesthetized with a mixture of ketamine (150 mg/kg) plus medetomidine (0.3 mg/kg), and a cohort of animals of each group was transcardially perfused with 0.9% saline followed by 4% paraformaldehyde. Brains were collected and transferred to a tube containing 4% PFA in PBS. After 48 h of incubation at room temperature (RT), brains were kept in 30% sucrose in PBS containing 0.01% of sodium azide. Afterward, brains were sectioned coronally on a vibratome (VT1000S, Leica, Wetzlar, Germany), and four series of coronal sections of striatum and SNpc at a thickness of 40 μm were selected for free-floating immunohistochemistry. Tyrosine-Hydroxylase (TH) detection was performed using the protocol described previously [30].

Then, TH+ labelling of dopaminergic fibers was performed in the dorsal striatum (6 slices/animal) and SNpc (4 slices/animal). Digital slides of the striatum were obtained using brightfield illumination (SZX16, Olympus, Tokyo, Japan) and the mean grey value was measured using the ImageJ software (v1.48, National Institute of Health, Stapleton, NY, USA) as previously described in detail [18]. Regarding SNpc, TH+ cells were visualized and counted using a stereological brightfield microscope (BX51, Olympus). The boundaries of the SNpc area were drawn, and total TH+ cells were counted in both hemispheres taking into account the whole nigral area. Data are presented as the percentage (%) of the contralateral side (intact side).

### 2.10. Statistical Analysis

A confidence interval of 95% was assumed for all statistical tests. The assumption of normality was tested for all continuous variables through evaluation of the frequency distribution histogram, the values of skewness and kurtosis, and through the Shapiro–Wilk test. The assumption of homoscedasticity was tested through Levene’s test. Both assumptions were met by all tested continuous variables. All continuous data are shown as the mean ± SEM. For the evaluation of mean differences in samples with one independent and one repeated measures variable, a mixed design ANOVA was carried out, with Tukey’s or Dunn’s post hoc test for pairwise comparison of the independent variable. For the comparison of means between two groups (rotameter test; discrete data), a Mann–Whitney U test was carried out. The statistical analysis and graphic representation were performed using the GraphPad Prism ver.8.0c (GraphPad Software; La Jolla, CA, USA). A statistical summary of all conducted analysis is presented in Appendix A.

## 3. Results

### 3.1. Generation of Human Midbrain-Specific Organoids

To develop an in vitro model relevant to PD neurodegeneration that is enriched in relevant cell populations (midbrain dopaminergic neurons) and also presents hallmark molecular features such as neuromelanin production, we used human midbrain-specific organoids (hMOS) from three different (hvNESCs) lines (Appendix A). An outline of the experimental paradigms for maintenance, differentiation, model induction, and secretome treatment of hMOS is presented in (Figure 1A). According to the schematic representation, during the first 12 days of culture, the hMOs rapidly increased in size and reached a mean core size of 1.25 mm (±0.19 mm, *n* = 5) in diameter after 20 days (Figure 1B). After 30 days in culture, some hMOS were collected to assess self-organization and neuronal differentiation.

We observed that hMOS present a stem cell niche in the center, and the dopaminergic neurons are distributed asymmetrically within the organoids (Figure 2a). Besides the observed robust differentiation into dopaminergic neurons, hMOs show the expression of midbrain floorplate markers FOXA2 and EN1 (Figure 2b). Additionally, we demonstrate the presence of dopamine (DA) and an evident co-expression of GIRK2 with TH, revealing the presence of the A9 subtype midbrain dopaminergic neurons (Figure 2c). These stainings revealed the formation of a complex neuronal network and are in line with the previously published results [15,16,17], demonstrating the specificity and reproducibility of these cultures.

During the development of the fetal human brain, neural-tube-derived cells not only differentiate into neurons but also into glia cells, including astrocytes and oligodendrocytes [31]. Thus, we also investigated the presence of astrocyte markers in the hMOS; however, no robust differentiation in astrocytes was observed at day 30 (we only observed some S100β+ and GFAP+ astrocytes in one specific location of the organoids; Appendix A). This is in agreement with brain development where glial differentiation temporally follows neuronal differentiation; therefore, a significant increase in astrocytes would be expected to be observed on more mature organoids (~day 60) as previously shown [15].

### 3.2. Neuroprotective Effects of BM-MSC Secretome on 6-OHDA-Induced Neurotoxicity in Human Midbrain-Specific Organoids

Afterwards, we used the hMO cultures as a platform to generate an in vitro PD model to study the neuroprotective effects of BM-MSC secretome. For that, hMOs were exposed to 6-OHDA (100 µM) for 2 days to induce dopaminergic degeneration, and then incubated for 6 days with total BM-MSC secretome (1×) immediately after 6-OHDA exposure (Figure 3). Importantly, a comparative analysis of the degree of 6-OHDA-induced degeneration in TH+ versus the overall mature MAP2 neuronal population demonstrated higher degeneration towards the TH+ population, highlighting the dopaminergic selectivity of our injury model (Appendix A). Then, we examined the effects of 6-OHDA and the respective treatments on the neuronal network within hMOs using image-based cell profiling. To probe the state of the overall neuronal populations after 6-OHDA treatments, we quantified the amount of the pan-neuronal marker TUJ-1+ neurons which remained unaltered (F (5,138) = 1.280, *p* = 0.235; Figure 4a). Due to TUJ-1 being a promiscuous marker for neuronal populations, as it labels both immature and mature neurons, we also assessed overall degeneration based on MAP2+ neurons which is expressed in more mature neurons [32]. In this regard, a significant decrease in MAP2+ neurons was observed after 6-OHDA exposure (F (5,138) = 10.22, *p* < 0.001; Figure 4b). We then looked specifically for TH+ dopaminergic neurons and both secretome and the basal medium (Control−) were able to maintain the cultures in the same conditions when compared to the organoids exposed to the differentiation medium (Control+). However, a significant decrease in TH+ cells was observed for the 6-OHDA-treated organoids (F (5,138) = 10.73, *p* < 0.001; Figure 4c), which validates the fitness for purpose of this optimized midbrain organoid system as a potential model for PD. Upon treatment, an almost significant trend (F (2,69) = 2.55, *p* = 0.08) in the increase in TH+ dopaminergic neurons was observed for 6-OHDA organoids treated with BM-MSC secretome when compared to the non-treated 6-OHDA organoids (Control−). 6-OHDA is a highly oxidizable dopamine analog, which can be captured through the dopamine transporters (DAT) [33]. As MAP2, DAT is a mature neuronal marker contrary to TUJ1, which is a marker of early committed neurons; therefore, we hypothesized that MAP2+ neurons could be more sensitive to 6-OHDA effects. We also computed a 3D mask for TH+ cells and generated a 3D skeleton of the dopaminergic network to extract features such as nodes (dendritic and axonal points of branching) and links (total number of branches), as well as neurite fragmentation (Appendix A and Appendix A). 6-OHDA treatment led to a decrease in the complexity of the dopaminergic neurons (links: F (5,138) = 33.56, *p* < 0.001; nodes: F (5,138) = 35.13, *p* < 0.001; Figure 4e,f) and an increase in fragmented neurites (F (5,138) = 10.87, *p* < 0.001; Figure 4d). Interestingly, we observed that the hMOS treated with BM-MSC secretome present significantly less fragmentation when compared to the non-treated group (F (2,69) = 6.17, *p* < 0.01; Figure 4d).

### 3.3. BM-MSC Secretome Reverts Parkinsons Disease Motor Symptomatology in a Unilateral Intrastriatal 6-OHDA Model

In this work, we addressed the effects of BM-MSC secretome using two different approaches: intracerebral (IC) injections into the lesioned areas (i.e., striatum and substantia nigra) and multiple intravenous (IV) injections for secretome administration. To establish an in vivo PD model, a total of 18 μg of 6-OHDA were injected at the right dorsolateral striatum, which selectively destroys dopaminergic neurons in the nigrostriatal pathway [34]. In this way, we were able to mimic the dopaminergic degeneration observed in PD on one side, while the other was used as a control of the lesion. Figure 5 outlines the in vivo experimental procedures.

### 3.4. Motor Behavior Analysis

Three weeks after lesion surgeries, animals were subjected to a baseline behavioral characterization at the motor level, including motor swimming and pole tests to evaluate motor coordination and strength, and the beam balance walk test to assess balance and fine motor coordination. The rotameter test was used to evaluate the imbalance in DA release between the denervated and the non-denervated striatum. All animals injected with 6-OHDA presented turning behavior after apomorphine administration, as revealed by the increased number of contralateral rotations of 6-OHDA-injected animals (Figure 6A), thereby confirming the success of the lesions on the ipsilateral side. Moreover, as observed in the first time point of graphs b-d of Figure 6, the 6-OHDA lesion caused a significant motor impairment phenotype in the sub-domains of motor coordination and loss of balance, as measured by the motor swimming, beam balance walk, and pole test paradigms. After performing the PD model characterization, 6-OHDA-lesioned animals were divided into four treatment groups: two groups received IC injections of Neurobasal A medium (IC Vehicle) or BM-MSC secretome (IC Secretome) in both dorsal striatum and SNpc, and the other two groups received multiple IV injections of Neurobasal A medium (IV Vehicle) or BM-MSC secretome (IV Secretome), as represented in Figure 5. Motor behavioral assessment was performed 1, 4, and 7 weeks after treatments. Results showed that IV Secretome-treated animals present a significantly better performance in motor paradigms in general (Figure 6B–D) in comparison with non-treated animals (IV Vehicle). Regarding the motor swimming test, this improvement was already visible 1 week after treatment administration and was sustained until 7 weeks after treatment (Figure 6B). Although not reaching statistical significance, the treatment with secretome via IC injections also improved the motor phenotype (mostly in the motor swimming and pole tests) of 6-OHDA mice, as it is possible to observe by its large effect size (see Appendix A). Of note, the route of secretome administration proved to be very important for motor function recovery, as the IV injections showed a relevant improvement on motor behavior, as well as a higher effect size, over the IC secretome administration. The results of the statistical analyses are presented in Appendix A.

### 3.5. Histological Analysis

To further analyze the effects of the 6-OHDA injections and the different treatments on the dopaminergic neuronal structure, histological analyses for TH were performed in both lesioned areas—dorsal striatum and SNpc (Figure 7A,C). The 6-OHDA injections caused a significant reduction in TH+ labelling in the dorsal striatum on the side of the lesion when compared to the Sham group (Figure 7B; raw data for Sham group of TH+ labelling/mm^2^—Contralateral side: 154.69 ± 17.01; Ipsilateral side: 141.52 ± 18.27). The same result was also observed after TH+ cell counts in the SNpc (Figure 7D; raw data for Sham group of TH+ cells/mm^2^—Contralateral side: 4.47 ± 0.58; Ipsilateral side: 4.18 ± 0.77). Both treatments with BM-MSC secretome (i.e., IC and IV injections) were able to significantly minimize the loss of dopaminergic fibers in the dorsal striatum (Figure 3B), which was not verified in the respective lesioned vehicle groups. Nonetheless, the IC Secretome was significantly better in this outcome when compared to the IV Secretome group. On the other hand, the SNpc results revealed that IC injection of BM-MSC secretome was able to significantly reduce dopaminergic cell loss in comparison to the non-treated group (Figure 7D). Nonetheless, although no statistical significance was reached between IV Secretome and the respective IV Vehicle, a large effect size was verified, suggesting an effect of this treatment approach in the neuroprotection of dopaminergic neurons in the SNpc. Histological analyses in the lesioned hemisphere were compared and presented as a percentage of the contralateral side, and the results of the statistical analyses are presented in Appendix A.

## 4. Discussion

In this work we have performed a translational investigation into the effects of BM-MSCs secretome for amelioration of Parkinson’s disease phenotypes using optimized human midbrain-specific organoids and a unilateral intrastriatal 6-OHDA mouse model of PD. Our in vitro modeling approach takes advantage of recent progress in cell culture technologies from the areas of embryology, stem cells, and neuroscience and aimed to produce an organoid system with higher predictive value for preclinical PD research. By using a guided approach to drive the differentiation of hvNESC, we were able to successfully induce ventral midbrain patterning and observe a robust presence of FOXA2 and EN1 floorplate markers. The differentiated organoids were enriched with a dopamine producing dopaminergic neuron population identified by strong TH+ and GIRK2+ co-labeling. These characteristics are desirable when modeling PD and the relative enrichment in this dopaminergic population is unparalleled when compared to unpatterned organoid models [15,16,17,35,36]. Due to the relevance of glial cells to PD pathophysiology [37], we also characterized the astroglial population contained within our organoid system. Although it is possible to witness the presence of cells stained for established astroglial markers (GFAP and S1000B), the labeling patterns are sparse and limited to only a few regions within the organoids. This suggests that the impact of astroglia as a relevant player for the instalment of PD phenotypes in our model is modest. Therefore, we centered our analysis on dopaminergic neurons. By using high-throughput confocal microscopy coupled with integrated 3D image analysis pipelines we demonstrated that 6-OHDA can be used to model key phenotypes of dopaminergic neuronal degeneration. For instance, significant changes in the cellularity of MAP2+ neurons within the organoids, as well as a specific and robust reduction in the TH+ population, were found when organoids were incubated with 100 μM 6-OHDA for two days. Importantly, although this temporal paradigm of 6-OHDA exposure induces degeneration of dopaminergic neurons, a significant number of neurons remain alive, phenocopying the natural differences in oxidative stress vulnerability witnessed in vivo at the substantia nigra [38,39].

Additionally, the profile of TH degeneration was accompanied by marked morphometric alterations (fragmentation and reduced complexity) of axons and neurites which together are markers of cytoskeletal disruption serving as a viable model to study restorative strategies [40,41,42,43]. In this context, our treatment with BM-MSC secretome was able to elicit near-significant protection specifically of the dopaminergic cell population (TH+). Moreover, when we assessed the morphometric alterations induced by 6-OHDA exposure, secretome significantly prevented the fragmentation of dopaminergic neuronal processes when compared to the control treated group. Together, these data demonstrate that secretome from BM-MSCs can protect dopaminergic neurons against 6-OHDA neurotoxicity in optimized mid-brain organoids. Furthermore, the temporal dynamics of our injury model indicate that the effect of the secretome may derive from pleiotropic mechanisms [9,10].

Significantly, although the organoids employed in this work are an unquestionable advancement over traditional 2D cell culture, they still present limitations. In general, current organoid systems lack two important aspects to conceptually form the basis of functional basal ganglia circuitry for the study of PD pathophysiology. The first aspect is the lack of microglial cells within brain organoids. These cells are crucial for orchestrating neuroimmune interactions such as the maturation of synaptic processes, as well as responses associated with cellular stress or injuries that are relevant to PD [44]. However, although promising, specific protocols need to be employed for the generation of microglia-enriched brain organoids, which is responsible for their limited usage to investigate PD-related phenotypes or therapeutic strategies [45,46]. The final caveat of current brain organoids is the relative difficulty of integrating functionally and anatomically separated brain circuits. In the context of PD, the interconnectivity between the substantia nigra and the striatum with cortical and thalamic structures underlie the functional impairments witnessed in animal models and human phenotypes. In order to conceptually contemplate these complex structures, assembloids of organoids that are grown and patterned from these different brain regions need to be created [47]. The feasibility of such an approach has been recently demonstrated but definite protocols for PD-specific contexts are still lacking [48]. After the proof-of-concept treatment effects from our injured hMOS demonstrated encouraging results, we performed an in vivo study of the effects of BM-MSCs secretome given via two distinct modes of administrations, two-site intracerebral or multiple intravenous injections. This work was also motivated by the need for a less invasive administration method to facilitate translation of earlier findings from our lab that collectively shows that BM-MSCs secretome exerts neurorestorative effects in PD [22,23,49,50].

For this, we established a unilateral model of dopaminergic degeneration based on the injection of 6-OHDA into the dorsolateral striatum of mice [29]. 6-OHDA-injected animals presented marked dopaminergic depletion, witnessed by the apomorphine test. The number of contralateral rotations following apomorphine administration suggests that injured animals had functional impairments in the nigrostriatal pathway, ascertaining both the construct and face-validity of our model [51,52]. Following the confirmation of our PD model induction and randomized treatments, we performed a temporal analysis of several motor behavior domains. The assessment of swimming ability measured by the time taken to cross a sixty-centimeter pool demonstrated that the BM-MSCs secretome injected intravenously preserved the injured animal’s motor abilities overtime. This effect was witnessed as early as one week following treatments and persisted throughout the end of the study. Although a smaller but similar trend was seen for the animals receiving IC secretome treatment, results failed to reach statistical significance. These results hold an important translational significance, as scores from swimming tests in mouse models of PD have been shown to be highly correlated with striatal dopamine levels [53,54]. IV injected secretome also induced gains in motor coordination and bradykinesia assessed by the pole test. The magnitude of the observed effects is noteworthy, with a near-total reversal of the diseased phenotype. In fact, when put in the perspective of similar therapeutic strategies for mouse models of PD, multiple intravenous administration of BM-MSCs secretome produced increased effects in the preservation of bradykinesia instalment [55,56,57]. The assessment of the animal’s ability to keep their balance while walking on a round beam, demonstrated that IV secretome significantly reduced the loss of balance after 6-OHDA injury. Interestingly, the performance of a similar beam-walking test to predict falls in Parkinson’s disease patients is currently being assessed on a multi-center clinical trial (NCT03532984). If the beam walk test proves to have high certainty to predict falls in humans, positive results in the test found in the preclinical setting with the use of therapeutic interventions will have increased predictive value [58]. Although only tendential, the high effect size from the IC secretome intervention points to a possible significant effect if studies with larger cohorts were to be conducted. Nevertheless, the differential effects observed with multiple IV secretome over the IC injections demonstrates the proof-of-concept and the superiority of the IV route. These improved effects may derive from a relative higher dosage of the neuroregulatory factors contained within the secretome, given the chronic administration regime, but also highlight a possible peripheral mode of action. To validate the findings from the motor behavior evaluation, we performed an immunohistochemical analysis of the patterns of dopaminergic degeneration (TH+ labeling) along the SNpc and the striatum. A differential response from the modes of administration was witnessed when the results from the motor behavior phenotypes are taken into consideration. In fact, the IC injection of secretome produced the highest effects in terms of dopaminergic neuron preservation at both the SNpc and the striatum when compared to its vehicle control, revealing a strong preservation of dopaminergic neurons and fibers. Interestingly, although the multiple IV injections of secretome elicited positive effects in the preservation of dopaminergic terminals in the striatum, results from the quantification of dopaminergic cells at the SNpc failed to reach statistical significance despite showing visible improvements when compared to its vehicle control. In our interpretation, these discrepancies highlight both the pathophysiological complexity of PD as well the intricacies in the relationship between the preservation of the nigrostriatal system and the improvements in motor symptomatology. In our extensive experience in modelling PD with 6-OHDA in rats, we find that intracerebral secretome elicits at least a two-fold increase in both measures of dopaminergic denervation (TH+ cell counts in the SNpc and fibers in the striatum) which historically correlates with functional improvements measured by skilled paw-reaching and rotarod tests [22,23,49,50]. In the current study, however, the baseline dopaminergic degeneration is subtler, rendering the effects of treatments smaller when compared to vehicle-treated controls in both routes of administration (IC and IV). In addition, in the current study, the relative (TH+ labeling) values reached in animals treated with IC secretome are considerably higher than the ones found in our previous works, even though these animals failed to significantly recover motor function. This may reveal a ceiling effect on the restoration of dopaminergic neurons to drive motor behavior improvements and highlight that other mechanisms may be relevant. Importantly, these discrepancies between the data from our previous works and the performance of IC secretome-treated animals in the present study, likely stem from differences in the injury protocols and the tests employed to assess motor function [23,29]. For instance, the degree of functional impairment induced by 6-OHDA injury depends not only on the dosage but also on the location and the spread of the toxin over multiple injection sites. Additionally, depending on the motor tests employed, the degree of emergence in motor phenotypes may differentially correlate with the severity of striatal dopaminergic denervation and nigral cell loss [59,60]. Overall, the results of the IC versus IV comparison revealed that the locally injected secretome (dorsal striatum and SNpc) induces a higher magnitude of protection of dopaminergic neurons. This result was not particularly surprising since intracerebral delivery circumvents blood–brain barrier (BBB)-associated resistance mechanisms and offers the possibility of reaching higher local substance concentrations in the appropriate target regions. Interestingly, although with IV secretome the magnitude of the effects on histological markers was smaller, the higher-magnitude effects observed in the motor function tests suggest that the modulation of non-dopaminergic pathophysiological mechanisms may be valuable targets for therapeutic success. In addition, the fact that observable histopathological gains are witnessed with peripheral secretome administration may highlight that the neurotrophic and immunomodulatory factors may be reaching the injured areas. Mechanistically, the positive results witnessed with the secretome treatments, both in our hMO system and at the histopathological and functional levels in injured animals, likely derive from the array of neuroregulatory molecules contained within the MSCs secretome. From proteomics investigations of the secretome of BM-MSCs, several proteins with potential therapeutic effects for PD are found [23,25,61]. Based on functional enrichment analysis of these proteomics datasets, together with in vitro and in vivo data from our lab, immunomodulatory, proteostasis and trophic support mechanisms are expected to be at least partly involved in the results herein observed. For instance, the secretome used in this study was able to reduce the expression of inducible nitric oxide from microglial cells exposed to an inflammatory stimuli [25]. This is of particular interest for our 6-OHDA mouse model, as we have demonstrated a striking glial reactivity suggestive of a persistent neuroinflammatory profile [29]. Furthermore, BM-MSCs secretome is able to promote neural differentiation of human neural progenitor cells and induce dopaminergic neuron survival in transgenic models of α-synuclein overexpression in *C. elegans*, demonstrating the ability to promote trophic support and enhance proteostasis mechanisms [62,63]. Although the provision of a specific mechanism of action for BMSC secretome in PD was not the aim of this study, we believe that our data are aligned with previous mechanistic studies employing components of the secretome as therapeutic strategies. For instance, we have demonstrated that the secretome has several proteins that are relevant for modulating PD-related pathophysiological mechanisms such as DJ-1 and MMP2 [23,25,61]. In this context, the presence of a DJ-1-based therapeutic approach has been shown to protect against the oxidative stress induced by 6-OHDA and MPTP in in vitro and rodent models of PD [64]. Furthermore, mechanistic investigations using BMSC secretome have demonstrated that the presence of MMP2 was responsible to reduce alpha-synuclein aggregates and protect dopaminergic neurons against MPTP injury, demonstrating a possible proteostatic effect [65]. Additionally, galectin-1, a positive regulator of notch signaling which is involved in neuronal protection and differentiation in the context of PD, was consistently found in our proteomic investigations [66,67]. Another possible mechanistic source that may explain the positive effects of the secretome in PD-related models is the action of small extracellular vesicles that possess both trophic and neuroprotective functions [68]. In fact, we have started to address this possibility by assessing the different fractions of BMSCs secretome (vesicular and soluble proteins) in a 6-OHDA rat model of PD. Our data demonstrated that there was no beneficial gain after fractionation, highlighting that the whole unfractionated secretome is superior in inducing functional and histological improvements [50]. In this study context, given the higher functional effects induced by the IV-secretome administrations, we are unable to rule out positive effects that can arise from the modulation of specific organ systems that may have indirect positive effects on the 6-OHDA-injured areas within the CNS. Of note, despite the existence of emerging studies demonstrating that secretomes from other cell sources can elicit beneficial effects in PD models [69,70], it is important to highlight that the use of secretomes from MSCs also circumvents several logistical constraints. For instance, the relative ease of isolation, maintenance, and fold expansion to numbers sufficient for clinical applications are key attributes that make MSCs superior over other cell types for regenerative applications. In summary, future studies aiming to address the mechanisms behind the positive functions of BMSC secretome in pre-clinical models of PD, namely the reliance on neuroprotection or neuroregeneration, will be instrumental in consolidating this therapeutic approach for the next steps towards clinical translation.

## 5. Conclusions

In this work, we employed different in vitro and in vivo PD models to test the secretome of BM-MSCs as a source of therapeutic factors with neuroprotective potential. We used a guided protocol approach to differentiate human neuroepithelial stem cells (hNESCs) into 3D human midbrain-specific organoids (hMOs). As a proof of concept, we produced toxin-induced dopaminergic neuronal cell death in the midbrain organoids using 6-hydroxydopamine (6-OHDA). We demonstrated that these organoids recapitulated key hallmarks of PD pathology, such as reduced amounts of midbrain dopaminergic neurons and neurite fragmentation. To the best of our knowledge, this is the first time that the BM-MSC secretome effects were explored using this advanced in vitro PD model, and we observed that the secretome was able to protect against the dopaminergic neuronal loss, as well as induced fragmentation of dopaminergic axons and neurites. Despite these promising results, future improvements can be performed on this model, not only to better dissect the mechanisms underlying dopaminergic cell death but also to evaluate additional molecular and functional readouts to understand which are the mechanisms behind the secretome effects. In a translational approach, we then used this neurotoxin model to further evaluate the effects of BM-MSC secretome, making a direct comparison between intracerebral injections (single administrations in the striatum and SNpc) with multiple systemic administrations. The BM-MSC secretome led to motor function recovery and dopaminergic loss protection; however, multiple systemic administrations showed a higher magnitude of therapeutic effect. This result holds clinical relevance because of the possibility for delivering MSC secretome in a non-invasive way, circumventing key hurdles associated with intracerebral injections. Important to note is the fact that we observed a stronger impact of secretome effects on the 6-OHDA in vivo model when compared to the 6-OHDA in vitro model using midbrain organoids, thus highlighting the need for further development of brain organoid technologies in order to model PD pathophysiological traits of higher complexity. Moreover, the systemic administration on the in vivo experiment had a higher impact on the animal’s motor recovery over the histological outcomes regarding dopaminergic neurons in the lesioned areas. Thus, we hypothesized that this differential neuroprotective outcome results from not only a direct but also an indirect effect of MSC secretome on dopaminergic neurons.

## Figures and Tables

**Figure 1 cells-12-02565-f001:**
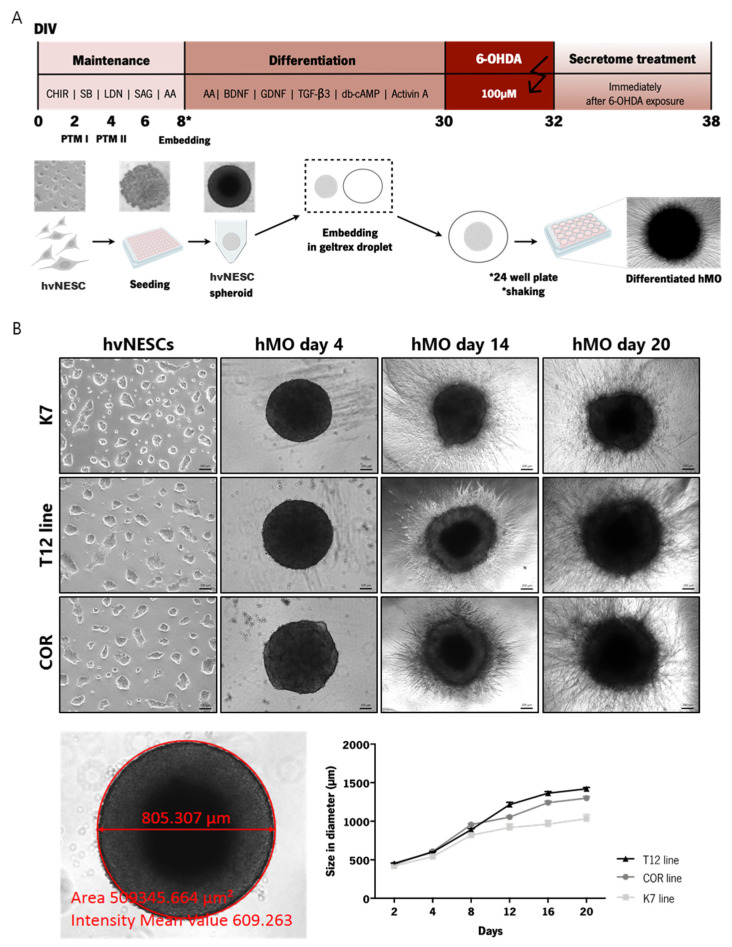
Derivation of hMOS from hvNESCs. (**A**) Schematic representation of the procedure of hMO culture system and the generation of the neurotoxin-induced PD model to assess the neuroprotective effects of BM-MSC secretome; and (**B**) growth of hMOs in culture: brightfield images of hMOS generated from hvNESCs at different days in culture (scale bar represents 100 and 200 μm), and the diameter size of hMOS per organoid line from day 2 to day 20. * Represents the specific time-point of geltrex embedding with plating and shaking in 24 well plates. Error bars represent mean ± SEM.

**Figure 2 cells-12-02565-f002:**
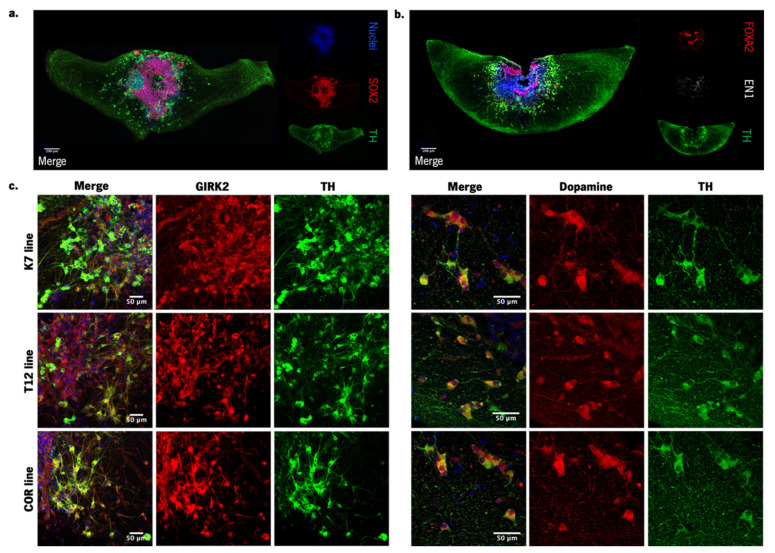
Characterization of hMOS at day 30 of the differentiation protocol. (**a**) hMO self-organization: stem cell niche in the middle (marked with SOX2 in red) and the dopaminergic neurons distributed asymmetrically (marked with TH at green) and nuclei (marked with DAPI in blue); (**b**) hMOs express midbrain floorplate markers like FOXA2 (red) and EN1 (gray) besides the characteristic TH (green) after 30 days in culture, and (**c**) express other midbrain markers besides TH such as GIRK2 (a marker of A9 neurons affected in PD) and dopamine, revealing the high specification for the midbrain, both stained in red.

**Figure 3 cells-12-02565-f003:**
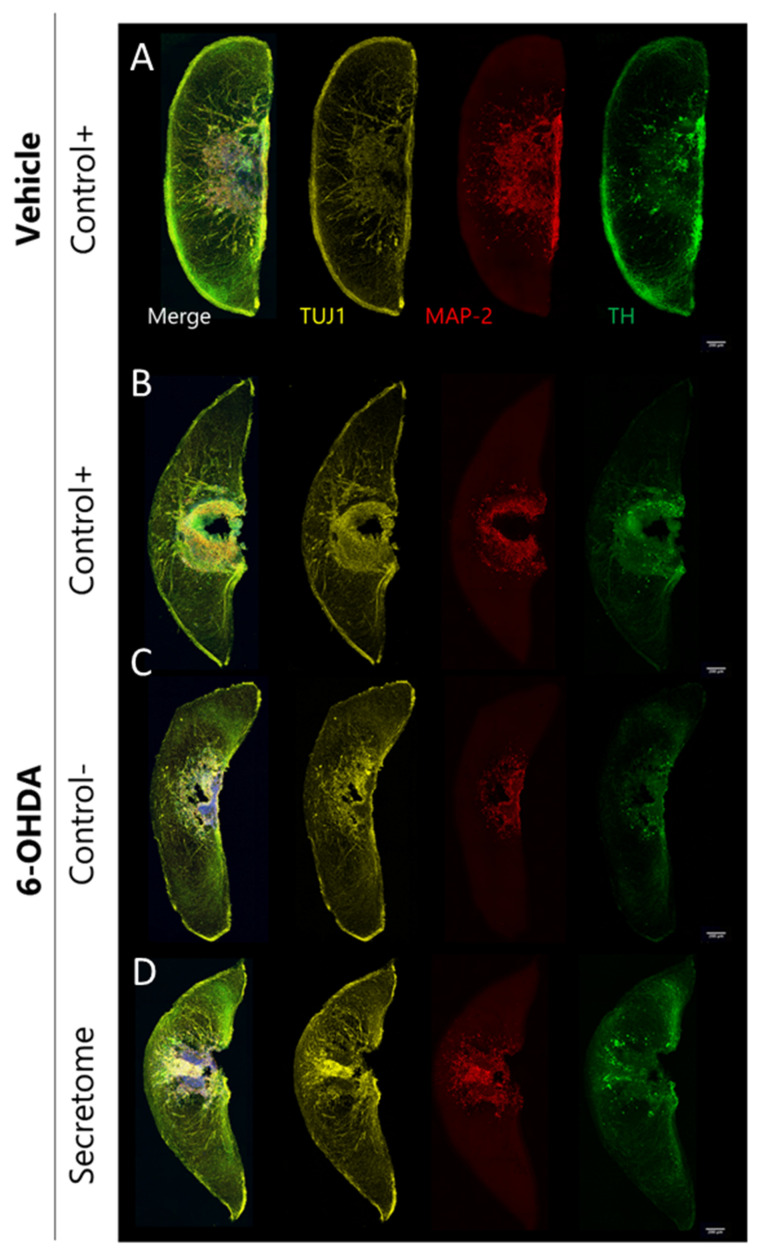
Representative images of the 6-OHDA-induced degeneration and BM-MSC secretome treatment effects. Representative maximum intensity projection of confocal images (20× magnification) of hMOS, showing TUJ1 (yellow), MAP2 (red), and TH (green) staining as well as merged images. (**A**) Representative organoids maintained in Control+ media without injury. (**B**) 6-OHDA-injured organoids for 48 h and treated with Control+ media. (**C**) 6-OHDA-injured organoids for 48 h and treated with Control- media. (**D**) 6-OHDA-injured organoids for 48 h treated with secretome.

**Figure 4 cells-12-02565-f004:**
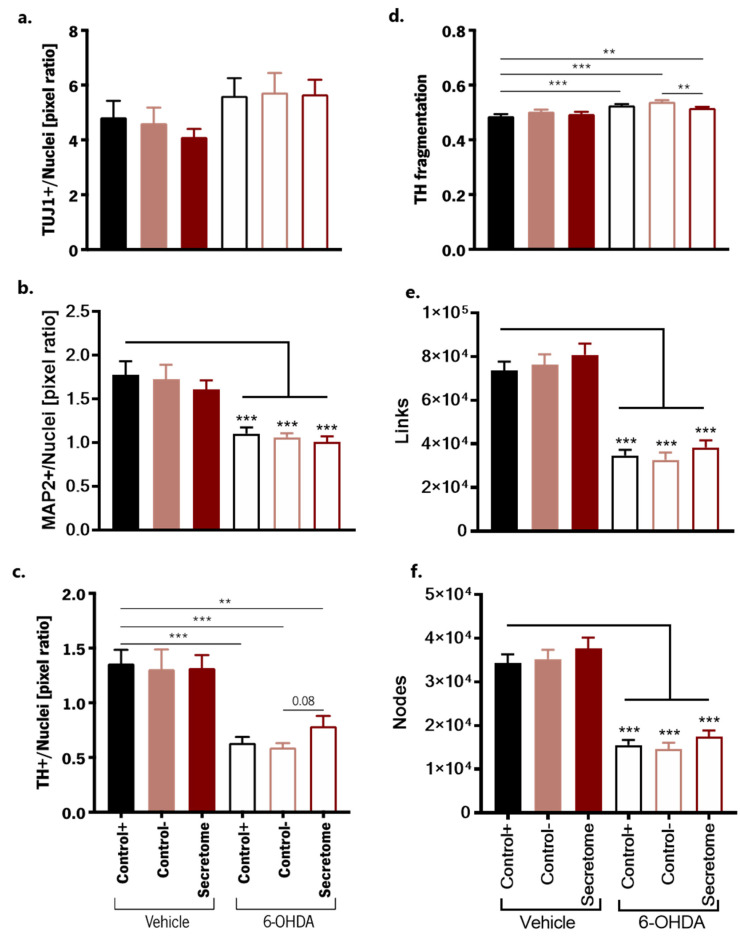
Effects of BM-MSC secretome in the 6-OHDA-induced PD model. Extracted features from high-content image analysis to evaluate the effects of 6-OHDA exposure and the respective treatments, namely in the overall amount of (**a**) TUJ+ neurons, (**b**) MAP2+ neurons, (**c**) TH+ dopaminergic neurons, and on neuronal complexity, namely (**d**) TH fragmentation, (**e**) links, and (**f**) nodes. Control+: Differentiation medium N2B27; Control− (a basal medium without any supplementation); Secretome (BM-MSC conditioned medium). Two independent experiments (24 sections analyzed/group). Data are presented as mean ± SEM. ** *p* < 0.01, *** *p* < 0.01.

**Figure 5 cells-12-02565-f005:**
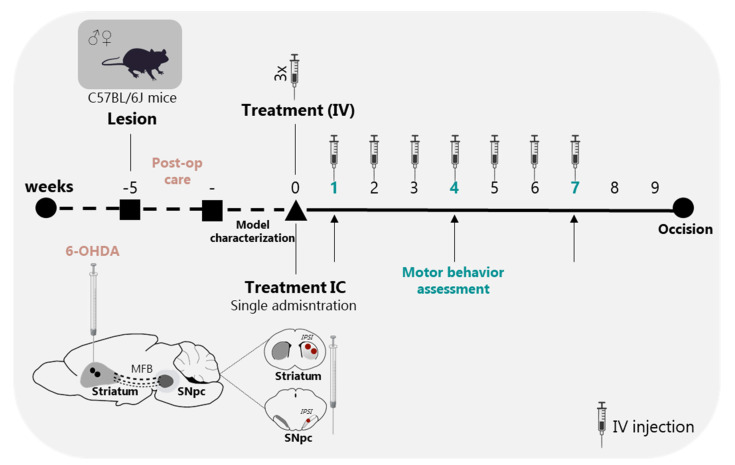
Experimental design. Schematic representation and temporal sequence of performed tasks throughout the in vivo experiment. Animals were injected with 6-OHDA into the dorsolateral striatum (2 injections); Mice were monitored daily for 14 days (post-operative care); 3 weeks after lesion, animals were characterized at motor level to validate the model; At week 5 post-lesion, animals were treated with BM-MSCs secretome using two different routes of administration: IC injections into the striatum and SNpc (2 injections into the striatum, and 1 injection into the SNpc—single administration) and repeated IV injections (3 injections in the first week, and 1 injection per week until the end of the experiment—10 injections in total). Motor behavioral assessment was performed 1, 4, and 7 weeks after treatments. Animals were sacrificed 12 weeks after lesion surgery and 9 weeks after treatments.

**Figure 6 cells-12-02565-f006:**
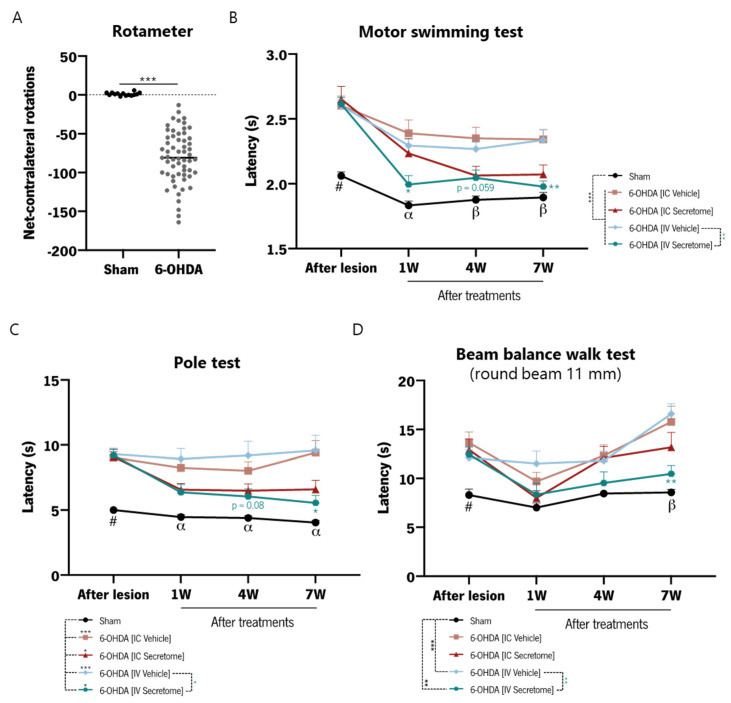
Motor behavioral analysis of 6-OHDA-lesioned animals after administration of BM-MSC secretome. (**A**) Rotameter test (apomorphine-induced rotations) as a measure of dopaminergic depletion induced by 6-OHDA lesion and compared to Sham animals; Motor performance of mice was evaluated using (**B**) motor swimming test, (**C**) pole test, and (**D**) beam balance walk test to assess the effects of the treatments. *n* = 13–15 for each group used. Statistical summary in Appendix A. Data are presented as mean ± SEM. * *p* < 0.05, ** *p* < 0.01, *** *p* < 0.001; #: Sham group statistically different from all groups, α: Sham group statistically different form all groups except from IV Secretome; β: Sham group statistically different form all groups except from IC Secretome and IV Secretome.

**Figure 7 cells-12-02565-f007:**
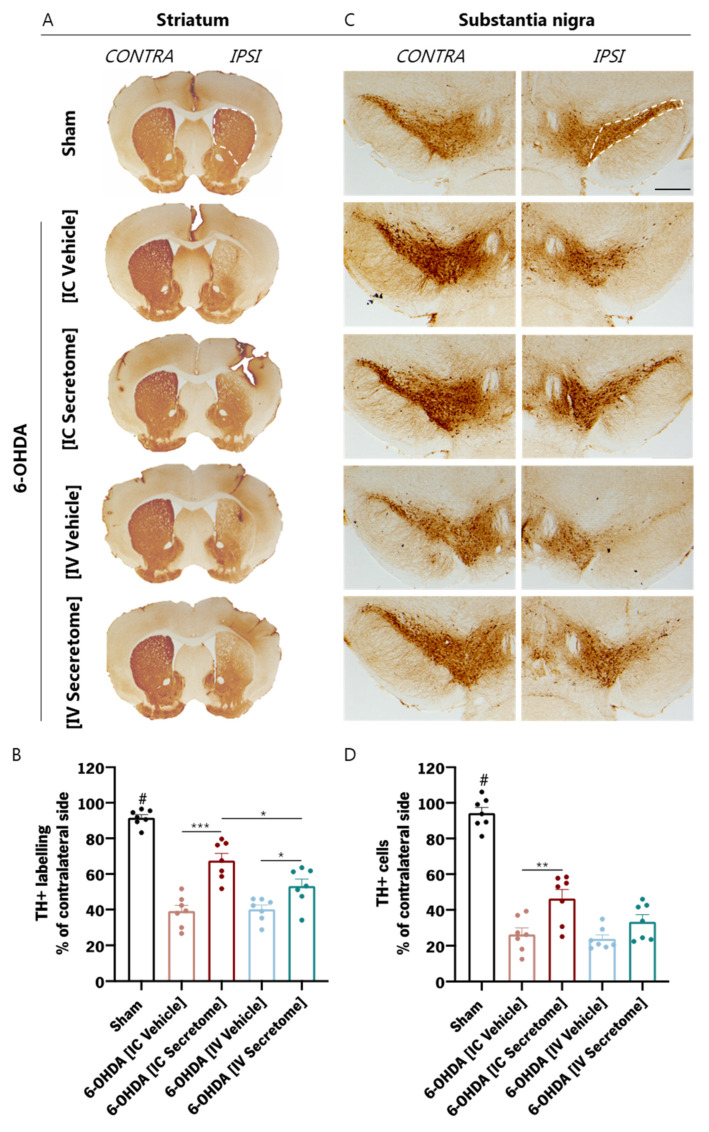
Histological analysis of the striatum and SNpc. (**A**) Representative images of the tyrosine hydroxylase (TH) staining on the striatum. (**B**) Quantification of TH+ labelling on the dorsal striatum, depicted as a percentage over the non-lesioned side. (**C**) Representative images of the TH staining on the ipsilateral and contralateral substantia nigra of the different experimental groups. (**D**) Quantification of TH+ cells on the SNpc, depicted as a percentage over the non-lesioned side. *n* = 7 for each group used. Statistical summary in Appendix A. Data are presented as mean ± SEM. * *p* < 0.05, ** *p* < 0.01, *** *p* < 0.001. # *p* < 0.001 from all other groups. Scale bar = 2 mm.

## Data Availability

Data are contained within the text and Appendix A.

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
