# Peer review of "Treating Parkinson’s Disease with Human Bone Marrow Mesenchymal Stem Cell Secretome: A Translational Investigation Using Human Brain Organoids and Different Routes of In Vivo Administration"

_cells, 2023, doi:10.3390/cells12212565_

Round 1

Reviewer 1 Report

The work of Mendes-Pinheiro et al. explores the therapeutic effect of the secretome of bone marrow-derived mesenchymal stem cells (BM-MSCs) in an in vitro model using human brain organoids, which recapitulated key hallmarks of PD pathology, and two routes (intracerebral administration or intravenous) of in vivo administration using a murine experimental model. With regard to in vitro results, treatment with BM-MSCs secretome was able to elicit a near significant protection specifically on the dopaminergic cell population. In addition, secretome significantly prevented the fragmentation of dopaminergic neuronal processes when compared to the control treated group. 

Interestingly, BM-MSCs secretome injected intravenously preserved the injured animal's motor abilities overtime in the in vivo study. Although the multiple IV injections of secretome elicited positive effects in the preservation of dopaminergic terminals in the striatum, results from the quantification of dopaminergic cells at the SNpc failed to reach statistical significance despite showing visible improvements when compared to its vehicle control. Even so, the results are very original and interesting. They open the possibility of future studies on the therapeutic potential of peripheral administration of the MSC secretome, thus avoiding both the limitations of cell therapy directly with MSCs and the drawbacks of intracerebral administration. However, there are several aspects that should be clarified: 

-Although the authors consider their in vitro results as a proof of concept that justifies the implementation of the in vivo model, it is not clear why different culture media are used to BM-MSC expansion before the secretome production for in vitro and in vivo studies. Can this variability condition a different potentiality of the secretomes? Why for the in vitro study, the secretome was 100× concentrated by centrifugation using a 5 kDa cut-off concentrator, but not for in vivo studies? 

-The authors should indicate the amount of protein administered intrathecally. 

-To my knowledge, the main limitation of the work is the absence of a reasonable explanation of the mechanism of neuroprotective action of the BM-MSCs secretome. The authors, in any case, should describe more explicitly the cocktail neurotophic proteins possibly involved in this action. On the other hand, it is known that extracellular vesicles derived from MSCs have neurotrophic effects in experimental models of various neurodegenerative diseases. Therefore, the authors should mention whether the BM-MSCs secretome that they used in the present study also contains these microparticles. And, if so, discuss their therapeutic advantages since they reproduce the effects of their parental MSCs, may have trophism towards inflammatory lesions, and, very importantly, cross the blood-brain barrier. 

Reviewer 2 Report

In their submitted manuscript Mendes-Pinheiro et al. investigate the potentially beneficial effects of the administration of human MSC-derived secretome to dopaminergic degeneration. The topic is very interesting, as currently there is no golden standard treatment for Parkinson's disease.

The study is straight-forward, the experiments have been planned, conducted, described and analyzed robustly and clearly. The results are promising; although, one of the main caveats of this study (as with numerous similar studies) is that we end up not really knowing what and how it has happened. Thus, the main message is that the MSC secretome is of interest.

Based on the above, it is my opinion that at least two issues have to be explored and noted in the discussion (which in its current form is very long and needs some trimming):

1. Since the behavioral and motor results are clear, do the authors exclude the possibility that they are mediated by systemic effects (e.g. metabolic)? how should that be explored? Does the higher effectiveness of the systemic administration point to that direction?

2. Do the authors beleive that the MSC-derived secretome is unique in some ways that can explain the results? If the secretome of other stem cells, or even differentiated cells, were to be used, what would the expectation be? 

The other issue that reduces the coherence of the manuscript is that the two parts (in vitro and in in vivo) are not linked in any rational way. It is obviously interesting to report the organoid protocol, but it would be nice to have something really connecting the two approaches.

In terms of the methods, my main point has to do with the fragmentation anlysis in the 6-OHDA treated organoids. Since Tuj-1 immunoreactivity remains unaltered and ther are MAP2 positive cells that are not TH+, I would like to see if fragmentation occurs in TH- cells. In other words, how sure are the authors that the pathology observed applies only or mostly to dopaminergic neurons; thus, allowing them to describe this as a PD in vitro model?

The final issue that needs to be discussed in a brief, but thoughtful way, is whether the data reflect neuroprotection or neuroregeneration. Is the study planned in a way that investigates only the former?  

Reviewer 3 Report

The research by Mendes-Pinheiro et al. focuses on investigating the potential benefits of using bone marrow-derived mesenchymal stem cell (BM-MSC) secretome to alleviate Parkinson's disease (PD) symptoms. To study this, they employed an innovative proof of concept approach by utilizing optimized human midbrain-specific organoids (hMOS) and a unilateral intrastriatal 6-OHDA mouse model of PD. hMOS exposed to 6OHDA presented dopaminergic neurons presented cytoskeletal disruptions, including fragmentation and reduced complexity of axon and neurites. Treatment with BM-MSC secretome showed promising results in protecting dopaminergic neurons from 6-OHDA-induced neurotoxicity, as it prevented both neuronal loss and morphological alterations. The study evaluated various motor behavior domains and found that intravenous administration of BM-MSCs secretome preserved the animals' motor abilities over time, including swimming ability, motor coordination, bradykinesia, and balance. On the other hand , The IC injection of secretome produced the highest effects in preserving dopaminergic neurons in both the substantia nigra pars compacta (SNpc) and the striatum when compared to its vehicle control. These findings hold translational significance and suggest the potential of secretome-based therapies for PD.

Minor comments:

1)    At a glance, it is challenging to understand Figure 3. It would be better if the authors broke it down into groups or provided a more detailed description in the figure legend.

2) Figure 4 b,e, and f statistic analysis is unclear. The authors should include a line connecting the tested or compared datasets.

Round 2

Reviewer 1 Report

This interesting work work was improved. Congratulations!

Reviewer 2 Report

In their responses to reviewers and in the revised manuscript Mendes-Pinheiro et al., have addressed most of the points raised.

Regarding the concerns I noted, I found the responses sucessful; nevertheless, I am surprised that none of these were included in the revised manuscript. As most of the points had to do with limitations of the study, I would strongly advise to include short references in the final text.

Figure 2 of the rebuttal letter should be added to the supplement.

A reference on the possible role of wider metabolic effects and an explanation on why MSC-derived secretome might be preferable over other secretomes, should be included in the discussion.
